# The Anti-Epileptic Drugs Lamotrigine and Valproic Acid Reduce the Cardiac Sodium Current

**DOI:** 10.3390/biomedicines11020477

**Published:** 2023-02-07

**Authors:** Lixia Jia, Arie O. Verkerk, Hanno L. Tan

**Affiliations:** 1Department of Clinical and Experimental Cardiology, Heart Center, Amsterdam UMC, University of Amsterdam, 1105 AZ Amsterdam, The Netherlands; 2Department of Medical Biology, Amsterdam UMC, University of Amsterdam, 1105 AZ Amsterdam, The Netherlands; 3Netherlands Heart Institute, 3501 DG Utrecht, The Netherlands

**Keywords:** anti-epileptic drugs, sudden cardiac death, cardiomyocytes, sodium current, action potentials

## Abstract

Anti-epileptic drugs (AEDs) are associated with increased risk of sudden cardiac death. To establish whether gabapentin, lamotrigine, levetiracetam, pregabalin, and valproic acid reduce the Na_v_1.5 current, we conducted whole-cell patch-clamp studies to study the effects of the five AEDs on currents of human cardiac Na_v_1.5 channels stably expressed in HEK293 cells, and on action potential (AP) properties of freshly isolated rabbit ventricular cardiomyocytes. Lamotrigine and valproic acid exhibited inhibitory effects on the Na_v_1.5 current in a concentration-dependent manner with an IC_50_ of 142 ± 36 and 2022 ± 25 µM for lamotrigine and valproic acid, respectively. In addition, these drugs caused a hyperpolarizing shift of steady-state inactivation and a delay in recovery from inactivation. The changes on the Na_v_1.5 properties were reflected by a reduction in AP upstroke velocity (43.0 ± 6.8% (lamotrigine) and 23.7 ± 10.6% (valproic acid) at 1 Hz) and AP amplitude; in contrast, AP duration was not changed. Gabapentin, levetiracetam, and pregabalin had no effect on the Na_v_1.5 current. Lamotrigine and valproic acid reduce the Na_v_1.5 current density and affect its gating properties, resulting in a decrease of the AP upstroke velocity. Gabapentin, levetiracetam, and pregabalin have no effects on the Na_v_1.5 current.

## 1. Introduction

Anti-epileptic drugs (AEDs) are a mainstay of epilepsy treatment and are also prescribed for behavioral problems and psychiatric disorders [1]. These drugs exert their anti-convulsant actions through various mechanisms, including the blocking of neuronal sodium channels [2]. Of clinical relevance, we and others found that AED use is associated with an increased risk of sudden cardiac death (SCD) due to cardiac arrhythmia [3,4]. The often-used drug carbamazepine is an example of such a drug [4]. We recently demonstrated that carbamazepine blocks the cardiac sodium channel Na_v_1.5 [4]. 

SCN5A encoded Na_v_1.5 is the most prominent sodium channel in the heart [5,6]. It is responsible for the rapid upstroke of the cardiac action potential (AP) and regulates impulse propagation in the heart. Na_v_1.5 block is a plausible mechanism contributing to the elevated SCD risk of cardiac arrhythmia and SCD of carbamazepine [7] because drugs that block the Na_v_1.5 increase SCD risk [8,9]. This insight was first gained in the landmark Cardiac Arrhythmia Suppression Trial in which patients randomized to the class 1c cardiac antiarrhythmic drugs (potent Na_v_1.5 blockers) flecainide or encainide suffered excess mortality rates due to SCD compared to placebo-treated patients [10]. On the other hand, Na_v_1.5 block by AEDs is plausible given that the Na_v_1.5 shares great homology with neuronal sodium channel isoforms [11,12]. 

In view of these observations, the aim of our study was to establish whether other AEDs than carbamazepine also block the Na_v_1.5. Here, we studied the five AEDs which, next to carbamazepine, have the largest number of users in the Netherlands, i.e., gabapentin, lamotrigine, levetiracetam, pregabalin, and valproic acid [13]. We conducted whole-cell patch-clamp studies to evaluate the effects of these drugs on the current densities and gating properties of Na_v_1.5 channels stably expressed in a HEK293 cell line, and on AP properties of freshly isolated rabbit cardiomyocytes.

## 2. Materials and Methods

### 2.1. HEK293 Cell Culture

We used a HEK293 cell line with stable human Na_v_1.5 channel expression [14]. The HEK293 cells were cultured in DMEM with Glutamax (Gibco), supplemented with 10% FBS (Biowest), L-glutamine (Gibco), penicillin-streptomycin (Gibco), and Zeocin (of 200 µg/mL, Invitrogen) in a 5% CO_2_ incubator (Shel Lab) at 37 °C. The cells were passaged every 3–4 d at 70% confluency in 25 mL flasks by using 0.25% trypsin (Gibco) treatments of around 2 min. On the day of patch-clamp measurements, cells were trypsined, stored at room temperature, and used within 3 h.

### 2.2. Rabbit Ventricular Myocyte Preparation

Male New Zealand white rabbits (3.0–3.5 kg) were anesthetized by a combination of ketamine (intramuscular 100 mg) and xylazine (intramuscular 20 mg), heparinized (Heparine LEO 5000 IU), and killed by an injection of pentobarbital (240 mg). Their hearts were excised and transported to the laboratory in cold (4 °C) Tyrode’s solution containing (in mM) 128 NaCl, 4.7 KCl, 1.5 CaCl_2_, 0.6 MgCl_2_, 27 NaHCO_3_, 0.4 Na_2_HPO_4_, and 11 glucose; pH 7.4 by equilibration with 95% O_2_ and 5% CO_2_. Subsequently, the hearts were mounted on a Langendorff perfusion apparatus and left ventricular midmyocardial myocytes were isolated by enzymatic dissociation from the most apical part of the left midmyocardial ventricular free wall, as described previously [15]. Animal procedures were performed in accordance with governmental and institutional guidelines for animal use in research and were approved by the Animal Experimental Committee of Amsterdam UMC, The Netherlands.

### 2.3. Patch-Clamp Recording

We applied the whole-cell configuration of the patch-clamp technique using an Axopatch 200B amplifier (Molecular Devices, San Jose, CA, USA). Borosilicate glass patch pipettes (GC100F-10; Harvard Apparatus, Cambourne, UK) had a resistance of 2–3 MΩ after filling with the pipette solution. All signals were low-pass filtered (5 kHz) and digitized at 33 kHz. Series resistance was compensated by ≥80%, and AP potentials were corrected for the calculated liquid junction potential [16] by an offline 15 mV shift in potential toward more negative values. In order to obtain steady-state conditions, signals were recorded after a stable stimulation period, i.e., under baseline conditions, and 5–7 min after the application of AEDs.

### 2.4. Sodium Current Measurements

The Na_v_1.5 current was measured in single HEK cells using a pipette solution containing (in mM) 10 NaF, 10 CsCl, 110 CsF, 11 EGTA, 1.0 CaCl_2_, 1.0 MgCl_2_, 2.0 Na_2_ATP, 10 HEPES (pH adjusted to 7.2 with CsOH), and a bath solution containing (in mM) 20 NaCl, 120 CsCl, 1.8 CaCl_2_, 1.0 MgCl_2_, 5.5 glucose, and 5.0 HEPES (pH adjusted to 7.4 with CsOH) [17]. The Na_v_1.5 current was measured at room temperature in response to depolarizing voltage steps from a holding potential of −120 mV (cycle length of 5 s). I_Na_ was defined as the difference between peak and steady-state current. The dose-response curves were fitted by the Hill equation: Y = 1/[(1 + (IC_50_/X)^n^)], where Y is the current normalized to baseline condition, IC_50_ is the dose required for 50% current block, and n is the Hill coefficient. The Na_v_1.5 (in)activation current was measured with a double-pulse protocol, as detailed in Section 3.2, below. During the first depolarizing pulses (P1), I_Na_ activates, and the currents analyzed here are used to determine current-voltage (I-V) relationships and the voltage dependency of activation. For the latter, the I-V relationships were corrected for driving force and normalized to maximum peak current. The second pulse (P2) is used to determine the voltage dependency of inactivation and currents were normalized to the largest I_Na_. Voltage dependence of activation and inactivation curves were fitted with the Boltzmann function: y = 1/{1 + exp [(V–V_1/2_)/k]}, where V_1/2_ is the midpoint of channel (in)activation, and k is the slope factor of the (in)activation curve. The use-dependent block was determined by application of 30 activating pulses from −120 to −20 mV at a frequency of 4 Hz, as detailed in Section 3.2. The Na_v_1.5 currents were normalized to the current of the first pulse. The length of recovery from inactivation was measured with a double-pulse protocol with two depolarizing steps (P1 and P2) from −120 to −20 mV and a variable interpulse interval (see Section 3.2). Currents measured during P2 were normalized to currents measured during P1. Recovery from inactivation was fitted by a double-exponential function: y = y0 + A_f_ [1–exp (−t/τ_f_)] + A_s_ [1–exp (−t/τ_s_)], where τ_f_ and τ_s_ are the fast and slow time constants of recovery from inactivation, respectively, and A_f_ and A_s_ are the fractions of fast and slow recovery from inactivation, respectively.

### 2.5. Action Potential Measurement

APs were measured in single rabbit ventricular cardiomyocytes using the amphotericin-perforated patch-clamp techniques at 36 °C. Cells were superfused with modified Tyrode’s solution containing (in mM) 140 NaCl, 5.4 KCl, 1.8 CaCl_2_, 1.0 MgCl_2_, 5.5 glucose, and 5.0 HEPES (pH adjusted to 7.4 with NaOH). Patch pipettes were filled with solution composed of (in mM) 125 K-gluconate, 20 KCl, 5.0 NaCl, 10 HEPES, and 0.44 Amphotericin-B (pH adjusted to 7.2 with KOH). APs were evoked at 1, 2, and 3 Hz using square 3-ms current pulses through the patch pipette. We analyzed resting membrane potential (RMP), AP amplitude (APA), maximal AP upstroke velocity (dV/dt_max_), and AP duration at 90% repolarization (APD_90_). AP parameters from 10 consecutive APs were averaged.

### 2.6. Preparation of Antiepileptic Drugs

All AEDs (purity ≥98%) were purchased from Sigma-Aldrich. Lamotrigine and valproic acid were dissolved in dimethyl sulfoxide (DMSO) to produce a 1 M stock solution. The stock solution was stored at −20 °C and freshly diluted in the bath solution to the desired concentration just before use. The concentration of DMSO in the final solution was less than 0.33% and this does not affect cardiac ion channels [18,19]. Pregabalin, gabapentin, and levetiracetam were freshly dissolved in the bath solution to the desired concentration just before use. The Na_v_1.5 current was measured at baseline conditions and after the wash-in of AEDs at concentrations of 1, 10, 30, 100, 300, or 1000 µM; these concentration ranges surrounded the therapeutic concentrations of the AEDs, as indicated by the yellow parts in Figure 1B–F [20,21,22,23,24,25,26,27].

### 2.7. Statistical Analysis

Values are presented as mean ± SEM. Curve fitting and statistics was performed using Prim8 GraphPad (GraphPad Software, San Diego, CA, USA). One-Way ANOVA or Two-Way ANOVA was used to assess the statistical significance of the differences among multiple groups. One-way repeated measures (RM) ANOVA was used, followed by pairwise comparison using the Holm-Sidak’s multiple comparisons test or One-way RM ANOVA on Ranks (Friedman test), followed by Dunn’s multiple comparison test for post hoc analyses when data were not normally distributed. Differences between the two groups were tested using paired Student’s t-tests or Two-Way RM ANOVA, followed by pairwise comparison using the Holm-Sidak’s multiple comparisons test. Details on normalization are given in Methods or in the figure legends. *p* < 0.05 was considered to be statistically significant.

## 3. Results

### 3.1. Inhibition of the Na_v_1.5 Current by Lamotrigine and Valproic Acid in a Concentration-Dependent Manner

First, we measured the effects of gabapentin, levetiracetam, pregabalin, lamotrigine, and valproic acid on the Na_v_1.5 current density in HEK293 cells. We applied 100 ms depolarizing pulses from −120 to −40 mV (Figure 1A) to HEK293 cell with stable Na_v_1.5 expression and tested various drug concentrations including the therapeutic concentrations, which are indicated as yellow parts in Figure 1B–F. We found that lamotrigine and valproic acid reduced the Na_v_1.5 current density in a concentration-dependent manner (Figure 1E,F). The average IC_50_ of lamotrigine and valproic acid were 142 ± 36 μM (*n* = 6–7) and 2022 ± 25 μM (*n* = 5), respectively. The tested concentrations of gabapentin, levetiracetam, and pregabalin had no effect on the Na_v_1.5 current density (Figure 1B–D).

### 3.2. Effects of Lamotrigine (100 μM) on Gating Properties of Na_v_1.5 Channels

Second, we tested if the decrease in Na_v_1.5 in response of 100 μM lamotrigine (close to IC_50_) is accompanied by changes in gating properties in HEK293 cells. Figure 2A shows typical Na_v_1.5 currents under baseline conditions and in the presence of 100 μM lamotrigine measured over a wide range of depolarizing voltages (for protocol, see Figure 2A, inset). The average current-voltage (I-V) relationships are shown in Figure 2B. Lamotrigine induced a similar amount of block over the whole voltage range measured and a hyperpolarizing shift in both activation and inactivation (Figure 2C). The V_1/2_ of channel activation occurred at −52.7 ± 1.7 mV in the absence, and −56.7 ± 1.5 mV in the presence, of lamotrigine (*n* = 9, *p* < 0.05). The V_1/2_ of channel inactivation were −92.9 ± 1.5 mV and −99.5 ± 2.6 mV, respectively (*n* = 9, *p* < 0.05, Table 1). The slope of the inactivation curve was significantly changed after the application of lamotrigine from 6.0 ± 0.4 to 7.7 ± 0.6 mV (n = 9, *p* < 0.05, Table 1). To study the rate-dependent effects of lamotrigine, we applied a double-pulse protocol with an interpulse interval of 50 ms (Figure 2D) and found that the reduction in the Na_v_1.5 current density at rising pulse numbers increased more in the presence of lamotrigine (Figure 2E). Consistent with this observation, this was attended by delayed recovery from steady-state inactivation (Figure 2E, Table 1) with τ_f_ and τ_s_ significantly changed from 11.2 ± 1.7 to 17.1 ± 3.6 ms, and from 134.8 ± 21.5 to 657.7 ± 125.1 ms, respectively (*n* = 7, *p* < 0.05, Table 1).

### 3.3. Effects of Valproic Acid (2000 μM) on Gating Properties of Na_v_1.5 Channels

Third, we studied the effects of 2000 μM valproic acid (close to IC_50_) on the Na_v_1.5 current in HEK293 cells, similar to that obtained for lamotrigine. Valproic acid reduced the Na_v_1.5 current density (Figure 3A,B). Figure 3C showed that valproic acid did not induce a statistically significant shift in voltage dependency of activation (V_1/2_ from −50.6 ± 1.5 to −50.5 ± 2.7 mV) (*n* = 15, *p* = NS) or slope of activation (Table 1). However, valproic acid induced a significant shift in steady-state inactivation (V_1/2_ from −92.7 ± 1.4 to −99.0 ± 1.9 mV) (*n* = 15, *p* < 0.05, Figure 3C, Table 1). And the slope of the inactivation curve was significantly changed after the application of valproic acid from 6.6 ± 0.2 to 5.9 ± 0.3 mV (*n* = 9, *p* < 0.05, Table 1). Valproic acid had modest effects on the rate of recovery from inactivation (Figure 3D,E, Table 1), affecting τ_f_ mildly (from 11.3 ± 2.1 to 17.5 ± 3.6 ms, *p* < 0.05), but not τ_s_ (from 165 ± 18.0 to 200.8 ± 34.5 ms, *n* = 7, *p* = NS). Accordingly, valproic acid significantly affected the rate of reduction of the Na_v_1.5 current density at repetitive pacing with an interpulse interval of 50 ms (*n* = 10, *p* < 0.05, Figure 3D).

### 3.4. Effects of Lamotrigine and Valproic acid on Action Potentials Properties

In a final patch clamp experiment, we studied the effects of lamotrigine (100 µM) and valproic acid (3000 μM) on APs elicited in rabbit ventricular cardiomyocytes in order to verify our findings regarding the effects of these AEDs on the Na_v_1.5 current in HEK293 cells, and to investigate possible drug effects on other ion channels. Figure 4A showed typical APs at 1 Hz under baseline conditions and in the presence of 100 µM lamotrigine; average AP parameters are summarized in in Figure 4B–D. Lamotrigine caused statistically significant decreases in dV/dt_max_ and APA in a frequency-dependent manner (Figure 4C,E). dV/dt_max_ was significantly decreased at all rates, e.g., by 43.0 ± 6.8% (from 309.7 ± 24.3 to 176.6 ± 19.0 V/s, *n* = 7, *p* < 0.05) at 1 Hz, and by 70.1 ± 8.2% (from 281.2 ± 29.0 to 84.0 ± 11.5 V/s, *n* = 7, *p* < 0.05) at 3 Hz (*n* = 7, *p* < 0.05) (Figure 4C). Similarly, APA decreased by 2.1 ± 0.5% (from 126.0 ± 1.3 to 123.4 ± 1.6 mV, *n* = 7, *p* < 0.05) at 1 Hz, and by 7.8 ± 1.1% (from 122.3 ± 1.9 to 112.8 ± 2.0 mV, *n* = 7, *p* < 0.05) at 3 Hz (Figure 4E). The reduction in dV/dt_max_ was larger at higher stimulation frequencies (*n* = 6–7, *p* < 0.05), consistent with the reduced rate of recovery from inactivation of Na_v_1.5 (Figure 2D,E). Lamotrigine did not change RMP or APD_90_. The absence of effects on RMP and APD_90_ indicates that other ionic currents were virtually unaffected.

Figure 5A shows typical APs under baseline conditions and in the presence of 3000 µM valproic acid at 1 Hz. The average AP parameters at 1 to 3 Hz are summarized in Figure 5B–E. Valproic acid also decreased dV/dt_max_ in a frequency-dependent manner, e.g., by 23.7 ± 10.6% (from 344.0 ± 20.6 to 261.9 ± 32.3 V/s, *n* = 10, *p* < 0.05) at 1 Hz, and by 37.4 ± 7.8% (253.6 ± 25.1 to 156.7 ± 38.0 V/s), *n* = 5, *p* < 0.05) at 3 Hz (Figure 5C). APA was also reduced, but only statistically significantly at 3 Hz (Figure 5E). There were no statistically significant effects on RMP or APD_90_, with the exception of a reduction in RMP at 3 Hz (Figure 5B,D).

## 4. Discussion

Our main findings were: (1) lamotrigine and valproic acid inhibited the Na_v_1.5 current in a dose-dependent manner, while gabapentin, levetiracetam, and pregabalin had no effect at the doses tested; (2) lamotrigine and valproic shifted the voltage dependency of inactivation and slowed the recovery from inactivation; (3) lamotrigine and valproic acid reduced dV/dt_max_ and APA in rabbit cardiomyocytes with a larger amount of reduction at fast pacing rates; and (4) lamotrigine and valproic acid did not impact other AP properties, except for modest reduction of RMP by valproic acid.

Our observations are largely consistent with reports on the effects of lamotrigine and valproic acid on neuronal sodium channels, reflecting the high homology between cardiac and neuronal sodium channels [11,12]. We observed a concentration-dependent blockade of the Na_v_1.5 current by lamotrigine, and similar effects were reported in both cardiac sodium channels expressed in HEK293 cells [28] and neuronal sodium channels expressed in CHO cells [29]. Meanwhile, lamotrigine reduced the density of voltage-gated sodium current in rat cerebellar granule cells and induced a hyperpolarizing shift in the voltage dependency of inactivation [30], which is consistent with our findings. Our observed lamotrigine-induced delay in recovery from inactivation is also mirrored by similar effects on cardiac sodium channels and neuronal sodium channels [29]. Previous reports on the effects of valproic acid on neuronal sodium channels were also consistent with our findings. For instance, valproic acid reduced sodium current density in the nodal membrane of peripheral nerve fibers of Xenopus laevis by 54% at a dose (2.4 mM) which is very close to our IC_50_ value of the Na_v_1.5 current block (2.0 mM) [31]. Moreover, valproic acid (2 mM) shifted the voltage dependence of inactivation to more hyperpolarized potentials in cortical neurons [32]. Furthermore, valproic acid (1 mM) reduced the sodium current density and slowed the recovery from inactivation in rat hippocampal neurons [33]. While some studies showed that valproic acid had no effect on fast neuronal sodium current, the concentrations used in these studies were mostly lower than therapeutic concentrations [34,35].

The antagonizing effects of lamotrigine and valproic acid on the Na_v_1.5 current, dV/dt_max_, and APA are here reported for concentrations (100 µM lamotrigine, 3000 µM valproic acid) that exceed the upper limit of the therapeutic range of lamotrigine (59 µM) and valproic acid (867 µM) [27], but only by a factor of 2–3 (while a reduction in sodium current density by lamotrigine already started at concentration within its therapeutic range). Thus, these effects may be clinically relevant because these somewhat elevated concentrations may occur in clinical practice, e.g., at (mild) overdoses. A drug-induced blockade of the human Na_v_1.5 current could reduce cardiac excitability (as evidenced by a widened QRS complex of the ECG) and increase mortality risk [36]. Accordingly, a review of case reports of lamotrigine overdose found that lamotrigine overdose may be associated with ECG changes (QRS widening) and cardiac arrhythmias (wide complex tachycardia, complete heart block) which are consistent with a reduced Na_v_1.5 current [37]. Another way in which our findings may have relevance in routine clinical care is that specific subgroups of patients may have elevated vulnerability to the Na_v_1.5 blocking effects of lamotrigine or valproic acid [37]. In these individuals, even plasma concentrations within the therapeutic ranges may cause clinically significant effects on cardiac electrophysiology, leading to cardiac arrhythmia and SCD. Enhanced vulnerability may stem from acquired comorbidities and/or from inherited susceptibility. The concept that acquired comorbidities may permit the occurrence of fatal cardiac arrhythmia and SCD upon use of Na_v_1.5 current blocking drugs was discovered in the Cardiac Arrhythmia Suppression Trial, in which patients randomized to the Na_v_1.5 current blockers flecainide or encainide suffered excess SCD rates compared to placebo-treated patients [10]. In a meta-analysis, it was discovered that this risk occurred in patients who have increased vulnerability to this adverse drug effect due to comorbidities associated with reduced Na_v_1.5 function, such as myocardial ischemia/infarction and heart failure [38]. This insight has prompted the recommendation in authoritative clinical guidelines to screen patients for the presence of these comorbidities and withhold these drugs from patients who have them [39]. On the other hand, inherited susceptibility may stem from carrying variants in genes that encode subunits of the Na_v_1.5 channel, in particular, variants in SCN5A, the gene that encodes its α-subunit. Loss-of-function mutations in this gene underlie the Brugada syndrome [40] and cardiac conduction disease [41], inherited cardiac arrhythmia syndromes associated with elevated SCD risk. Accordingly, mutations in SCN5A were found in a series of patients with epilepsy who suffered unexplained and autopsy-negative SCD (sudden unexplained death in epilepsy), and these mutant genes, when heterologously expressed in CHO-K1 cells, produced altered Na_v_1.5 channel functional properties [42]. Of interest, one of these patients used lamotrigine at the time of SCD [43]. In view of these observations, when prescription of lamotrigine or valproic acid is considered, it could be prudent to first investigate whether the patient has any acquired or inherited condition that would increase the vulnerability to excessive Na_v_1.5 channel block which could culminate in SCD. This strategy would mirror the strategies in routine cardiology practice to screen patients on these conditions before cardiac drugs that block Na_v_1.5 channels (e.g., flecainide or other class I antiarrhythmic drugs) are considered [39], and to withhold these drugs from patients with Brugada syndrome or those who carry SCN5A mutations [44]. Screening for inherited vulnerability may be facilitated by the rapidly increasing availability of widespread DNA testing. Before full implementation of DNA testing, inquiring about the presence of familial SCD during simple history taking may already be informative because of the familial nature of SCD [45]. Conversely, the use of lamotrigine and valproic acid may not confer increased risk of cardiac arrhythmias and SCD in patients without enhanced vulnerability.

In any case, when we consider our observations in view of previous pharmacoepidemiologic studies into the associations between AED use and the risk of SCD [3,46,47], we conclude that blocking effects on Na_v_1.5 channels do not fully account for the increased risk of SCD associated with epilepsy [48]. Our conclusion derives from the fact that our observed effects on Na_v_1.5 currents were only partly consistent with the findings in these pharmacoepidemiologic studies. For instance, while we report that lamotrigine blocks Na_v_1.5 currents, this drug was not associated with increased SCD risk in a study of Eroglu et al. [46]. While the comparator in that study was valproic acid, the SCD risk of lamotrigine actually tended to be smaller (but statistical significance was not reached). In the studies of Bardai et al. [46] and Hookana et al. [47], the numbers were too small to draw conclusions on possible effects of lamotrigine on SCD risk. A recent review also found that there is not sufficient evidence to support or refute the notion that lamotrigine is associated with increased SCD risk [49]. For valproic acid, increased SCD risk was reported by Hookana et al., but not by Bardai et al. (a possible effect on SCD risk was not studied by Eroglu et al., who used valproic acid as comparator). Conversely, while we found that gabapentin, levetiracetam, and pregabalin had no effects on Na_v_1.5 currents, Eroglu et al. found that pregabalin conferred higher SCD risk than valproic acid, while gabapentin and levetiracetam also tended to have higher SCD risk (but it was not statistically significantly). Similarly, Bardai et al. reported higher SCD risk for gabapentin (for levetiracetam, the statistical power was too small to draw meaningful conclusions). Still, Hookana et al. reported no elevated SCD risk for gabapentin and pregabalin. We conclude from these comparisons that other mechanisms beyond the Na_v_1.5 block also contribute to the elevation in SCD risk in epilepsy, as previously reported [50].

## 5. Conclusions

Lamotrigine and valproic acid reduce the Na_v_1.5 current by reducing its current density and changing its gating properties; these effects are reflected in changes in AP properties. Gabapentin, levetiracetam, and pregabalin have no effects on the Na_v_1.5 current.

## Figures and Tables

**Figure 1 biomedicines-11-00477-f001:**
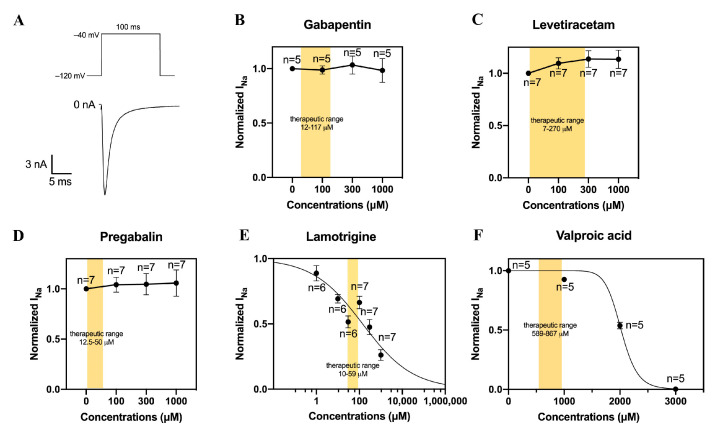
Effects of anti-epileptic drugs on current densities of Na_v_1.5 channels expressed in HEK293 cells. (**A**) Typical Na_v_1.5 current elicited by application of 100 ms depolarizing pulses from −120 to −40 mV. (**B**–**D**) Normalized Na_v_1.5 current in the absence or presence of 100, 300, or 1000 µM gabapentin (**B**), levetiracetam (**C**), and pregabalin (**D**). (**E**,**F**) Lamotrigine and valproic acid induced concentration-dependent inhibition of the Na_v_1.5 current magnitude. Solid lines are Hill fits to the average data. Values are normalized to the values measured under baseline conditions. The yellow parts indicate the therapeutic concentrations of the AEDs. Numbers near symbols indicated the number of cells (n) measured at the given concentrations.

**Figure 2 biomedicines-11-00477-f002:**
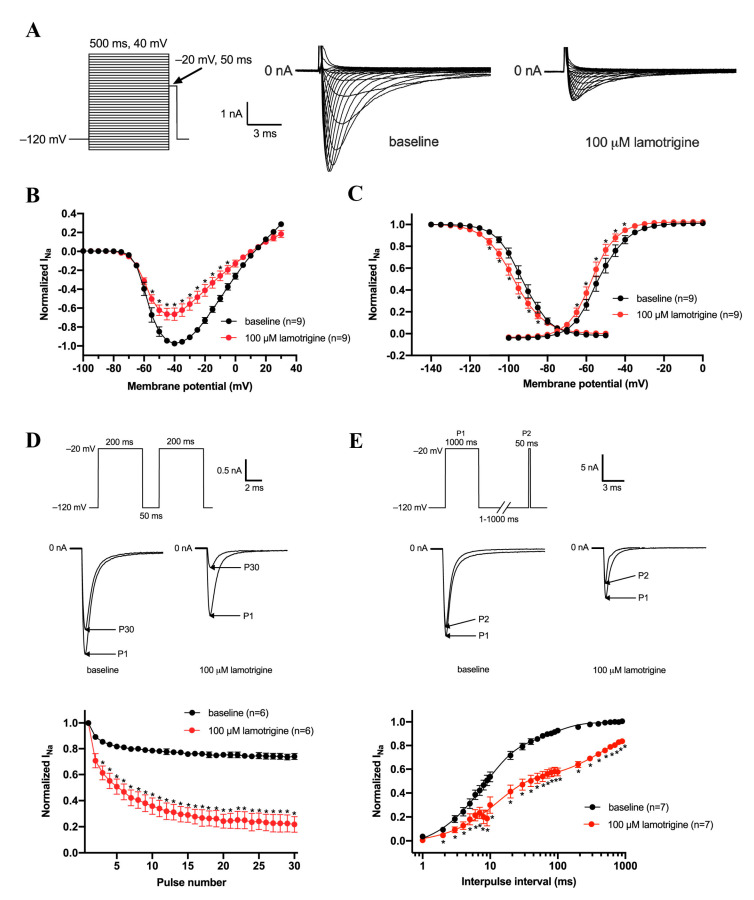
Effects of 100 µM lamotrigine on density and gating kinetics of Na_v_1.5 channels expressed in HEK293 cells. (**A**) Typical Na_v_1.5 currents under baseline conditions and in the presence of 100 μM lamotrigine. Inset: voltage clamp protocol used to measure current-voltage (I-V) relationships (**B**) and the voltage dependency of (in)activation (**C**). Cycle length was 5 s. (**B**) The Na_v_1.5 current-voltage (I-V) relationships before and after the application of 100 μM lamotrigine. The Na_v_1.5 current was normalized to the maximal peak amplitude under baseline conditions, but the peak current was set to −1 to retain the well-known inward direction of sodium current. (**C**) Effects of lamotrigine on the voltage dependency of the Na_v_1.5 current (in)activation. Solid lines are Boltzmann fits to the average data. (**D**) Use dependency under baseline conditions and in the presence of lamotrigine measured during a train of 30 depolarizing pulses with an interpulse interval of 50 ms. Inset: voltage clamp protocol used to measure (**upper panel**) and typical Na_v_1.5 currents under baseline conditions and in the presence of 100 μM lamotrigine (**middle panel**). (**E**) Recovery from inactivation of the Na_v_1.5 current in the absence and presence of 100 µM lamotrigine measured with a double-pulse protocol with variable interpulse intervals. Inset: voltage clamp protocol used to measure (**upper panel**) and typical Na_v_1.5 currents under baseline conditions and in the presence of 100 μM lamotrigine with an interpulse interval of 50 ms (**middle panel**). * *p* < 0.05 lamotrigine versus baseline (Two-Way RM ANOVA).

**Figure 3 biomedicines-11-00477-f003:**
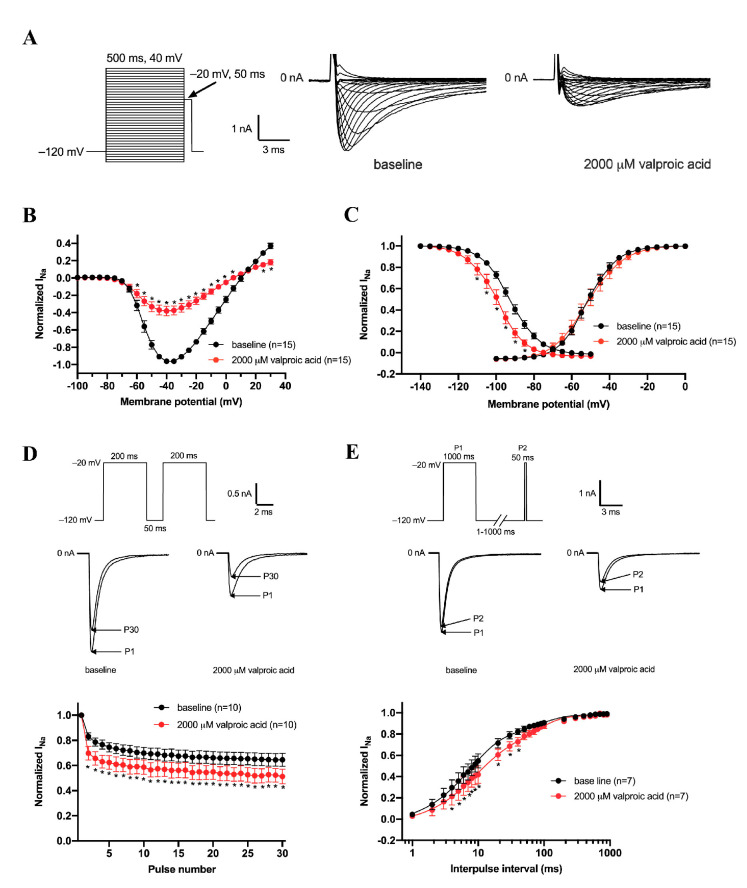
Effects of 2000 µM valproic acid on the density and kinetics of currents from Na_v_1.5 channels expressed in HEK293 cells. (**A**) Typical Na_v_1.5 under baseline conditions and in the presence of 2000 μM valproic acid. Inset: used voltage clamp protocol to measure I-V relationships (**B**) and the voltage dependency of (in)activation (**C**). Cycle length was 5 s. (**B**) Average I-V relationships before and after the application of 2000 μM valproic acid. (**C)** Effects of valproic acid on Na_v_1.5 (in)activation. Solid lines are Boltzmann fits to the average data. (**D**) Use dependency under baseline conditions and in the presence of valproic acid. Inset: voltage clamp protocol used to measure (**upper panel**) and typical Na_v_1.5 currents under baseline conditions and in the presence of 2000 μM valproic acid (**middle panel**). (**E**) Recovery from inactivation of the Na_v_1.5 current in the absence and presence of 2000 μM valproic acid measured with a double-pulse protocol with variable interpulse intervals. Inset: voltage clamp protocol used to measure (**upper panel**) and typical Na_v_1.5 currents under baseline conditions and in the presence of 2000 μM valproic acid with an interpulse interval of 50 ms (**middle panel**). * *p* < 0.05 valproic acid versus baseline (Two-Way RM ANOVA).

**Figure 4 biomedicines-11-00477-f004:**
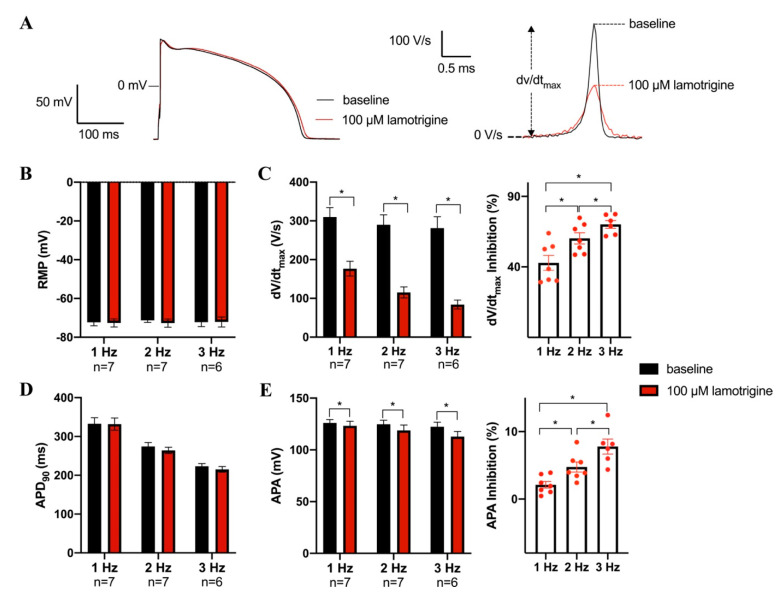
Effects of 100 µM lamotrigine on action potentials (APs) of rabbit ventricular cardiomyocytes. (**A**) Superimposed representative APs at 1 Hz under baseline conditions and in the presence of 100 µM lamotrigine. Inset: time derivatives of the AP upstrokes. The dV/dt_max_ are aligned by the peak. (**B**–**E**) Average AP characteristics at 1, 2, and 3 Hz of resting membrane potential (RMP, (**B**)), maximal AP upstroke velocity (dV/dt_max_, (**C**, **left panel**)) and the inhibition percentage of dV/dt_max_ induced by lamotrigine (**C**, **right panel**), AP duration at 90% of repolarization (APD_90_, (**D**)), AP amplitude (APA, (**E**, **left panel**)), and the inhibition percentage of APA induced by lamotrigine (**E**, **right panel**). Data are mean ± SEM. Numbers below frequencies indicate the number of cells (n) measured at a given frequency. * *p* < 0.05 lamotrigine versus baseline (Two-Way RM ANOVA or One-Way RM ANOVA).

**Figure 5 biomedicines-11-00477-f005:**
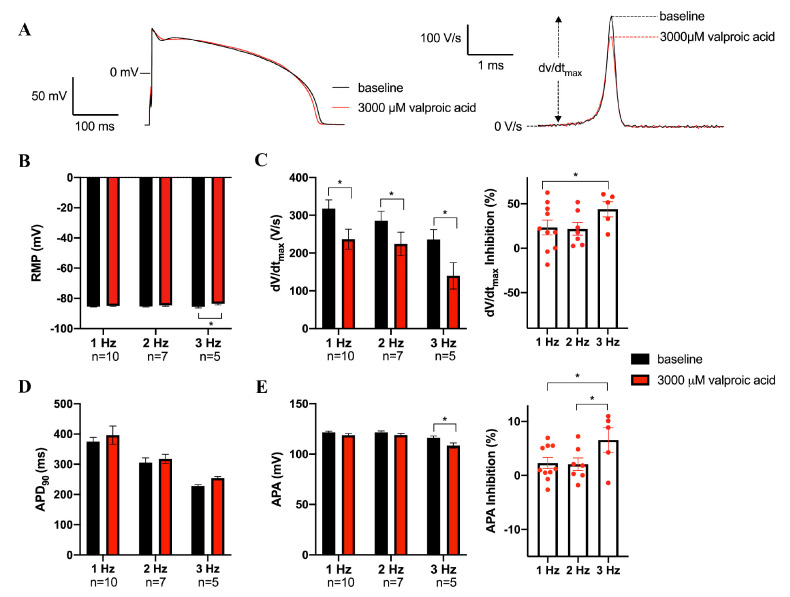
Effects of valproic acid on APs of rabbit ventricular cardiomyocytes. (**A**) Superimposed representative APs at 1 Hz under baseline conditions and in the presence of 3000 µM valproic acid. Inset: time derivatives of the AP upstrokes. The dV/dt_max_ are aligned by the peak. (**B**–**E**) Average AP characteristics at 1, 2, and 3 Hz of resting membrane potential (RMP, (**B**)), maximal AP upstroke velocity (dV/dt_max_, (**C**, **left panel**)), the inhibition percentage of dV/dt_max_ induced by valproic acid (**C**, **right panel**), AP duration at 90% of repolarization (APD_90_, (**D**)), AP amplitude (APA, (**E**, **left panel**)), and the inhibition percentage of APA induced by valproic acid (**E**, **right panel**). Data are mean ± SEM. Numbers below frequencies indicate the number of cells (n) measured at a given frequency. * *p* < 0.05 valproic acid versus baseline (Two-Way RM ANOVA or One-Way RM ANOVA).

**Table 1 biomedicines-11-00477-t001:** Cardiac sodium current properties in the absence (baseline) and presence of 100 µM lamotrigine and 2000 µM valproic acid.

		Activation		Inactivation		Recovery from Inactivation		
		V_1/2_ (mV)	k (mV)	V_1/2_ (mV)	k (mV)	τ_f_ (ms)	τ_s_ (ms)	A_s_/(A_s_ + A_f_) (ms)
lamotrigine	baseline	−52.7 ± 1.7 (*n* = 9)	5.7 ± 0.4 (*n* = 9)	−92.9 ± 1.5 (*n* = 9)	6.0 ± 0.4 (*n* = 9)	11.2 ± 1.7 (*n* = 7)	134.8 ± 21.5 (*n* = 7)	0.18 ± 0.02 (*n* = 7)
	wash-in	−56.7 ± 1.5 * (*n* = 9)	5.1 ± 0.3 (*n* = 9)	−99.5 ± 2.6 * (*n* = 9)	7.7 ± 0.6 * (*n* = 9)	17.1 ± 3.6 * (*n* = 7)	657.7 ± 125.1 * (*n* = 7)	0.44 ± 0.03 * (*n* = 7)
valproic acid	baseline	−50.6 ± 1.5 (*n* = 15)	5.9 ± 0.3 (*n* = 15)	−92.7 ± 1.4 (*n* = 15)	6.6 ± 0.2 (*n* = 15)	11.3 ± 2.1 (*n* = 7)	165.0 ± 18.0 (*n* = 7)	0.16 ± 0.02 (*n* = 7)
	wash-in	−50.5 ± 2.7 (*n* = 15)	5.8 ± 0.3 (*n* = 15)	−99.0 ± 1.9 * (*n* = 15)	5.9 ± 0.3 * (*n* = 15)	17.5 ± 3.6 * (*n* = 7)	200.8 ± 34.5 (*n* = 7)	0.2 ± 0.02 (*n* = 7)

V_1/2_, membrane potential for half-maximal (in)activation; k, slope factor of (in)activation curve; τ_f_ and τ_s_, are fast and slow time constant of recovery from inactivation, respectively; and A_f_ and A_s_, fractions of fast and slow recovery from inactivation, respectively. Data are expressed as the mean ± SEM. baseline vs AEDs, * *p* < 0.05 (paired Student’s *t*-tests).

## Data Availability

Data available in a publicly accessible repository.

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
