# Peer review of "The Anti-Epileptic Drugs Lamotrigine and Valproic Acid Reduce the Cardiac Sodium Current"

_biomedicines, 2023, doi:10.3390/biomedicines11020477_

Round 1

Reviewer 1 Report

In this paper, the authors investigated the effect of anti-epileptic drugs altering the biophysical properties of cardiac sodium current. The motivation came from the discovery that anti-epileptic drugs has been associated with increased risk of sudden cardiac death. They investigated the effects of the most used anti-epileptic drugs: gabapentin, lamotrigine, levetiracetam, pregabalin and valproic acid. Patch-clamp recordings were performed in HEK293 cell culture and rabbit ventricular myocyte. The authors found that lamotrigine and valproic acid change the sodium current properties.

The results of this paper provide a foundation to future research of the side effects of anti-epileptic drugs based on their ability to modulate sodium channel properties.

In my opinion, patch-clamp recordings were well executed and the conclusions are well supported by the data. However, I have certain concerns that I will mention below:

Minor points:

The statistical analysis of the data showed in Fig 5E related to the APA inhibition (%) seems to be odd. Visual comparison of the groups 1 Hz vs 2 Hz suggests that they are not significantly different. Both groups show similar mean values and similar data spread.

I suggest to the author to double check that.

Author Response

We thank the reviewer for his/her time and efforts to review our manuscript and his/her constructive comments. We took the reviewer’s comments to heart and made changes to the manuscript accordingly. Our response to each of the reviewer’s specific comments is given below, repeating the reviewer’s comment in bold, followed by our response. Changes made to the manuscript are detailed here and appear in the revised manuscript as ‘tracked changes’ through the ‘Track Changes’ function of MS Word, as requested by the editors.

In this paper, the authors investigated the effect of anti-epileptic drugs altering the biophysical properties of cardiac sodium current. The motivation came from the discovery that anti-epileptic drugs has been associated with increased risk of sudden cardiac death. They investigated the effects of the most used anti-epileptic drugs: gabapentin, lamotrigine, levetiracetam, pregabalin and valproic acid. Patch-clamp recordings were performed in HEK293 cell culture and rabbit ventricular myocyte. The authors found that lamotrigine and valproic acid change the sodium current properties.

The results of this paper provide a foundation to future research of the side effects of anti-epileptic drugs based on their ability to modulate sodium channel properties.

In my opinion, patch-clamp recordings were well executed and the conclusions are well supported by the data. However, I have certain concerns that I will mention below:

Minor points:

 The statistical analysis of the data showed in Fig 5E related to the APA inhibition (%) seems to be odd. Visual comparison of the groups 1 Hz vs 2 Hz suggests that they are not significantly different. Both groups show similar mean values and similar data spread.

 I suggest to the author to double check that.

Response 1: We thank the reviewer for pointing to this issue. We apologize for having mislabeled the Figure. We have double checked and confirm that the groups 1 Hz vs 2 Hz in this Figure are not significantly different, but that the groups 1 Hz vs 3 Hz are significantly different.  We have corrected this in Figure 5E of the revised manuscript.

Reviewer 2 Report

The article is interesting and presents a high amount of work. Thus, I think the article should be published after resolving the following points:

1 – I think it's a bit abusive to obtain results in kidney of a human embryo and compare them with cardiomyocytes, I don't think a direct comparison should be made.

2- What do you mean by this sentence “potentials were corrected for the calculated liquid junction potential” how was this correction made?

3 - The concentration of DMSO in the final solution was less than 0.33% in pacth clamp? at this concentration DMSO kills the cell? please show an original tracing showing current recovery.

Author Response

We thank the reviewer for his/her time and efforts to review our manuscript and his/her constructive comments. We took the reviewer’s comments to heart and made changes to the manuscript accordingly. Our response to each of the reviewer’s specific comments is given below, repeating the reviewer’s comment in bold, followed by our response. Changes made to the manuscript are detailed here and appear in the revised manuscript as ‘tracked changes’ through the ‘Track Changes’ function of MS Word, as requested by the editors.

The article is interesting and presents a high amount of work. Thus, I think the article should be published after resolving the following points:

1 – I think it's a bit abusive to obtain results in kidney of a human embryo and compare them with cardiomyocytes, I don't think a direct comparison should be made.

Response 1: We thank the reviewer for her/his suggestion. In electrophysiological studies, cell expression systems such as the HEK293 cell line are frequently used to study the biophysical properties of various ion channels.  In the present study, we used HEK293 cells with stable human NaV1.5 channel expression which allowed us to assess the effects of gabapentin, lamotrigine, levetiracetam, pregabalin, and valproic acid in detail on current density and gating properties of NaV1.5 current. Although cell expression systems are proven to be valuable tools in ion channel studies, they do not have a cardiomyocyte background. Therefore, we set out to confirm our principal findings observed in HEK293 cells by doing experiments in freshly isolated cardiomyocytes. The results of these studies were highly consistent with our findings in HEK293 cells.

2- What do you mean by this sentence “potentials were corrected for the calculated liquid junction potential” how was this correction made?

 Response 2: We apologize for being unclear. The action potentials were offline corrected for the calculated 15 mV liquid junction potential by a 15 mV shift of the potentials towards more negative values. We have adapted the sentence, which now reads: “……, and AP potentials were corrected for the calculated liquid junction potential [16] by an offline 15 mV shift in potential toward more negative values.”

3 - The concentration of DMSO in the final solution was less than 0.33% in pacth clamp? at this concentration DMSO kills the cell? please show an original tracing showing current recovery.

Response 3: DMSO may affect ion channel function if used at too high concentrations [1]. Our used concentration of maximal 0.33% does not affect sodium currents as previously reported [2,3]. We have included this information in the methods section on page 3, which now reads: “The concentration of DMSO in the final solution was less than 0.33% and this does not affect cardiac ion channels [18,19]".

We followed the reviewer’s suggestion and have now included typical examples of use dependency and the current recovery (Fig 2D, Fig 2E, Fig 3D, Fig 3E).

References

[1] Ogura T, Shuba LM, McDonald TF. Action potentials, ionic currents and cell water in guinea pig ventricular preparations exposed to dimethyl sulfoxide. Journal of Pharmacology and Experimental Therapeutics 273.3 (1995): 1273-1286.

[2] Matsuki NO, Quandt FN, Ten Eick RE, Yeh JZ. Characterization of the block of sodium channels by phenytoin in mouse neuroblastoma cells. Journal of Pharmacology and Experimental Therapeutics 228.2 (1984): 523-530.

[3] Theile JW, Cummins TR. Inhibition of Navβ4 peptide-mediated resurgent sodium currents in Nav1. 7 channels by carbamazepine, riluzole, and anandamide. Molecular Pharmacology 80.4 (2011): 724-734.

Round 2

Reviewer 2 Report

This article was revised appropriately.

I recommend accept